# Removal of Antimony(V) from Drinking Water Using nZVI/AC: Optimization of Batch and Fix Bed Conditions

**DOI:** 10.3390/toxics9100266

**Published:** 2021-10-14

**Authors:** Huijie Zhu, Qiang Huang, Shuai Fu, Xiuji Zhang, Zhe Yang, Jianhong Lu, Bo Liu, Mingyan Shi, Junjie Zhang, Xiaoping Wen, Junlong Li

**Affiliations:** 1Henan International Joint Laboratory of New Civil Engineering Structure, College of Civil Engineering, Luoyang Institute of Science and Technology, Luoyang 471023, China; huijiezhu@lit.edu.cn (H.Z.); hqwhy@foxmail.com (Q.H.); shuaifu16@163.com (S.F.); zhangxiuji1987@163.com (X.Z.); kaiyzhey17947@126.com (Z.Y.); junjie194312@163.com (J.Z.); wenxiaoping053@163.com (X.W.); lijunlong6868@163.com (J.L.); 2School of Environmental and Municipal Engineering, North China University of Water Resources and Electric Power (NCWU), Zhengzhou 450046, China; lujianhong@ncwu.edu.cn; 3College of Civil Engineering, Guangzhou University, Guangzhou 510006, China; shmygz@gzhu.edu.cn; 4Laboratory of Functional Molecular and Materials, School of Physics and Optoelectronic Engineering, Shandong University of Technology, Zibo 255000, China

**Keywords:** active carbon-supported nano-zero-valent iron(nZVI/AC), Antimonate(Sb(V)), adsorption, modeling, mechanism

## Abstract

Antimony (Sb) traces in water pose a serious threat to human health due to their negative effects. In this work, nanoscale zero-valent iron (Fe^0^) supported on activated carbon (nZVI) was employed for eliminating Sb(V) from the drinking water. To better understand the overall process, the effects of several experimental variables, including pH, dissolved oxygen (DO), coexisting ions, and adsorption kinetics on the removal of Sb(V) from the SW were investigated by employing fixed-bed column runs or batch-adsorption methods. A pH of 4.5 and 72 h of equilibrium time were found to be the ideal conditions for drinking water. The presence of phosphate (${\mathrm{PO}}_{4}^{3-}$), silicate (${\mathrm{SiO}}_{4}^{2-}$), chromate (${\mathrm{CrO}}_{4}^{2-}$) and arsenate (${\mathrm{AsO}}_{4}^{3-}$) significantly decreased the rate of Sb(V) removal, while humic acid and other anions exhibited a negligible effect. The capacity for Sb(V) uptake decreased from 6.665 to 2.433 mg when the flow rate was increased from 5 to 10 mL·min^−1^. The dynamic adsorption penetration curves of Sb(V) were 116.4% and 144.1% with the weak magnetic field (WMF) in fixed-bed column runs. Considering the removal rate of Sb(V), reusability, operability, no release of Sb(V) after being incorporated into the iron (hydr)oxides structure, it can be concluded that WMF coupled with ZVI would be an effective Sb(V) immobilization technology for drinking water.

## 1. Introduction

Antimony (Sb) is the tenth most mined metal in the world, with yearly production totaling more than 1.0 × 10^5^ tons [1]. The majority of the world’s active Sb mines are situated within China, the world’s most prominent Sb producer, accounting for almost 90% of the global output of 187,000 tons per year [2]. Plastic catalysts, flame retardants, pigments, batteries, glass, and ceramics are just a few of the industries that employ Sb. It has, however, been proven to expose significant threats to human health and ecosystems [3,4,5,6,7,8,9,10,11,12].

Antimony, like arsenic, has an s^2^p^3^ outermost orbital electronic distribution and so has a total of two valence states, Sb(III) and Sb(V), which are the most common Sb species found in the environment. Over the range of environmentally significant pH, Sb(V) mainly occurs in the form of ${\mathrm{Sb}(\mathrm{OH})}_{6}^{-}$ under aerobic conditions [2]. ${\mathrm{Sb}(\mathrm{OH})}_{6}^{-}$, an octahedral framework has a sizeable ionic radius and low density of charge, with several differences from P(V) and As(V), leading to weaker binding on solid surfaces than P(V) and As(V) [13]. On account of its neutral character over an extensive range of pH, as well as its low solubility, ${\mathrm{Sb}(\mathrm{OH})}_{6}^{-}$ is more mobile in alkaline and neutral environments [14]. As a result, Sb(V) is chosen as the target contaminant in this work. In naturally occurring waters, it is found in the form of oxyanions, and the majority of inorganic antimony is found as ${\mathrm{Sb}(\mathrm{OH})}_{6}^{-}$. In the range of natural pH, Sb(V) mostly occurs as ${\mathrm{Sb}(\mathrm{OH})}_{6}^{-}\;$ and may be stable across a large redox range [2].

China, as well as the US Environmental Protection Agency (EPA) and the European Union (EU), have identified several Sb-related chemicals as priority pollutants. The maximum Sb contamination level is currently 5 μg·L^−1^, as mandated by China’s newly released National Drinking Water Standard [15]. The EU, as well as the US EPA, have now designated it as a priority contaminant. In water employed for drinking purposes, Sb content should be below 6 μg·L^−1^, a limit set by the US EPA. In China, the comparable limit for Sb content is 5 g·L^−1^, which is the same as the EU and WHO guidelines. Antimony concentrations in non-polluted waterways are typically less than 5 μg·L^−1^. Nonetheless, it could reach as high as 53.6 ± 46.7 μg·L^−1^ in a water body near multiple antimony-related mining deposits.

Bioremediation, reverse osmosis, ion exchange, coagulation (coprecipitation), and adsorption are some of the current approaches for eliminating Sb from water [16,17,18,19,20,21,22]. Owing to technological and cost advantages, adsorption is predominantly utilized for removing Sb from drinking water. Iron (hydr)oxides and elemental iron are commonly used as adsorbents for Sb removal due to their strong affinity for Sb [1,23,24,25]. Adsorption is widely acknowledged as a very successful, cost-efficient, and safe approach. A large number of adsorbents with a special affinity for Sb have been produced to date. Zero valent iron (ZVI) has been widely used to treat water contaminated with metalloids and heavy metals in recent years [16,26,27,28,29,30,31]). ZVI has a significant Sb removal effectiveness and functions via a variety of modes of interaction, such as precipitation, adsorption, coprecipitation, and size-exclusion, in addition to its inexpensive cost and easy availability. As a consequence of its huge active surface area and strong Sb adsorption capacity, nanoscale zero-valent iron (nZVI) was recently identified as a suitable candidate for the elimination of Sb from drinking water [32,33]. Zero-valent iron and iron (hydr)oxides, on the other hand, are usually fine powders that, unless granular, cannot be used in fixed-bed columns. For water treatment operations involving fixed beds, ferric hydroxide could potentially be processed as a granular substance [34] or put onto polymeric anion exchangers [35], sand, and other materials. Due to its small particle size, the direct use of nZVI in water treatment systems could lead to iron contamination and rapid nZVI loss in drinking water. As a result, in order to treat Sb-contaminated drinking water, nZVI must be loaded onto supporting materials [34,36]. We recently impregnated nZVI nanoparticles with activated carbon (AC) to create a new hybrid adsorbent in our research. The resulting compound removed arsenic from the aqueous solution quite well [37]. In addition, some low-cost minerals, such as sand and diatomite, could function as hosts for nZVI. A weak magnetic field (WMF) can promote ZVI corrosion, resulting in faster Fe^2+^ release and more sequestration of Sb(III), Sb(V) As(V), As(III), Se(IV), and Cu(II). This is because the WMF could reduce Sb(V) passivation of ZVI and thus promote Sb(V) removal by corroded nZVI. [12,13,38]. Therefore, in this study, we proposed the utilization of ZVI in combination with WMF to achieve effective immobilization of Sb(V).

The goal of this work was to prepare supported nZVI and assess its ability to remove Sb from water. Because of its superior mechanical strength and porous architectures, AC was chosen to function as a supporting material (though its adsorption capacity for Sb is less than that of nZVI). The effects of several parameters on Sb elimination were investigated, including adsorbent dose, pH, dissolved oxygen (DO), and common ions.

## 2. Materials and Methods

### 2.1. Chemical and Instrumentation

All the analytical grade chemical reagents used in this study were purchased from Sinopharm chemical reagent Co., Ltd. (Sinopharm chemical reagent Co., Ltd., Shanghai, China). All solutions were produced by utilizing deionized (DI) water. The stock solution of Sb(V) (100.0 mg·L^−1^) used in our investigations was produced from K_2_H_2_Sb_2_O_7_·4H_2_O.

### 2.2. Synthesis and Characterization of nZVI/AC

The methods of synthesizing and characterizing nZVI/AC can be seen from reference [37]. The main features of nZVI/AC are given in Table 1.

### 2.3. Batch Adsorption Experiments

The ordinary AC’s Sb(V) elimination was 0.93% and overlooked in the study.

To characterize the factors influencing the sorption process, a series of experiments were performed by employing a 100 mL solution of Sb(V) with an initial concentration of 1.0 mg·L^−1^ and 1.5 g·L^−1^ nZVI/AC. The mass of nZVI/AC (0.5, 1.0, 1.5 and 2.0 mg·L^−1^), the solution pH (pH 3, 5, 7, 9 and 11), and adsorption kinetics were all investigated in this regard (0–72 h). Following the addition of nZVI/AC to the solution, the pH was brought to neutral with diluted NaOH or HCl. The solution was allowed to filter via a 0.22 μm membrane filter for aqueous Sb(V) measurement after 72 h of equilibration at pH 6.5 and 25 ± 1 °C in a shaker.

At pH 6.5 at 25 ± 1 °C, the influence of humic acid and anions (such as phosphate, silicate, sulphate, carbonate, oxalate) on the adsorption of Sb(V) upon nZVI/AC was examined by contacting 100 mL of 1.0 mg·L^−1^ solution with the adsorbent of 1.5 g for 72 h. The additional ions to arsenic molar ratio was 10:1, while 5 mg·L^−1^ of humic acid was added.

When investigating the influence of high dissolved oxygen (DO_H_), high-purity oxygen (O_2_ > 99.9%) was introduced for >30 min in the forward solution where nZVI/AC was added. When the anoxic state was simulated, the forward solution where nano-zero valent iron was added was passed through. High-purity nitrogen (N_2_ > 99.9%) was introduced for >30 min to eliminate DO in the solution.

The effect of common anions (phosphate, silicate, sulfate, carbonate, oxalate), cations (ferrous, calcium, magnesium) and humic acid on the adsorption of arsenic onto the NZVI/AC was investigated at pH 6.5 at 25 °C by mixing 0.15 g of adsorbent with 100 mL of 1.0 mg·L^−1^ solution for 72 h. The molar ratio of the added ions to arsenic was 10:1 and the concentration of humic acid was set as 5 mg/L.

An atomic fluorescence spectrophotometer (AFS-2202E, Haiguang Corp., Beijing, China) and a hydride generator were used to quantify the content of Sb(V) present in the solution. The instrument’s detection limit was 0.1 µg·L^−1^. The concentration of residual in the adsorbent (q_t_, mg·g^−1^) was determined using the following expression
(1)$$q_{t}=\frac{{V(C}_{0}{\;-\;C}_{t})}{W_{s}}$$

The volume of solution (mL) is given by V, the initial concentration is given by C_0_ and C_t_ represents the Sb(V) at a time t (in mg·L^−1^). The adsorbent weight (g) is given by W_s_. The percentage of removed Sb(V) (R%) is calculated making use of the below-mentioned expression:(2)$$R\;\left(\%\right)=\frac{C_{0}-{\;C}_{t}}{C_{0}}\;\times \;100$$

### 2.4. Fixed Bed Column Study

Column experiments were performed to see if WMF could be used to improve nZVI reactivity in a continuous flow system. Two parallel continuous column systems possessing an inner diameter of 10 mm (outer diameter = 14 mm) and a height of 200 mm were set up, as shown in Figure 1. One column system had a series of ring-shaped permanent magnets with a maximum value of magnetic field flux intensity as ~1800 Gs (Material: NdFeB, inner diameter = 14 mm, outer diameter = 25 mm, thickness = 5 mm, Shenzhen Min Magnetic Technology Co., Ltd., Shenzhen, China), with a ring magnet every 5.0 cm. A mixture of 10.62 g of nZVI/AC was used to fill the polymethyl methacrylate (PMMA) columns (10–20 mesh). Both ends of the column were plugged with a small piece of glass wool. Figure 1 depicts the experimental setup of the column and the parts that go with it. The experiments were performed at 25 ± 1°C. To achieve an appropriate flow rate, a peristaltic pump (HL-1S, Shanghai Huxi Analytical Instrument Co., Shanghai, China) was utilized. The Sb(V) ion solution was transferred upward via the nZVI/AC beds to guarantee that the bed was saturated perfectly. To assess the conservation of Sb(V), effluent samples were taken at regular intervals. The experiments were carried out until the adsorbents were saturated, after which the column data was analyzed.

The column’s exhausted nZVI/AC was removed and stored in a 250 mL flask for regeneration. The suspension was shaken at 150 rpm for 5 h at 25 ± 1 °C with NaOH (0.5 mol·L^−1^) that is equal to 5 times of nZVI/AC volume, followed by the removal of alkaline solution and the process was repeated four times, and de-ionized water was employed for the elution of nZVI/AC exposed to alkaline solutions.

### 2.5. Desorption of Adsorbed Sb(V)

After proceeding with the reaction of the adsorbent (0.15 g) with 100 mL of 2 mg·L^−1^ Sb(V) for 72 h, the resulting antimony-loaded nZVI/AC was rinsed using pre-distilled water to eliminate leftover Sb(V) solution. The nZVI/AC was agitated for 12 h after being combined with 100 mL of NaOH 0.1 M at pH 13.

### 2.6. Models

In the current study, the kinetics of Sb(V) adsorption were calculated making use of an intraparticle diffusion model and the below-mentioned expression [39].
q_t_ = k_id_ t^0.5^(3)
q_t_ (mg·g^−1^) denotes the amount of Sb(V) adsorbed at t time, k_id_ represents the original intraparticular diffusion rate (mg·g^−1^·h^−0.5^).

### 2.7. Equilibrium Uptake Studies

For a certain flow rate and inlet concentration, the maximal value of column capacity, q_total_ (mg), equals the area under the plot of the adsorbed Sb(V) concentration C_ad_ (C_ad_=C_0_−C_e_, where C_0_ is influent metal ions concentration) (mg·L^−1^) versus time (h) and C_e_ is effluent metal ions concentration and is calculated as mentioned below [40]:(4)$$q_{\mathrm{total}}=\frac{\mathrm{QA}}{1000}=\frac{Q}{1000}\int_{t=0}^{{t=t}_{\mathrm{total}}}C_{\mathrm{ad}}\mathrm{dt}$$

In the above expression, the flow rate (mL·min^−1^), the total flow time (min), and the area under the breakthrough curve are indicated by Q, t_total_, Q is, and A, respectively. The equilibrium uptake (q_eq_(exp)), the weight of Sb(V) adsorbed per unit dry weight of adsorbent(mg·g^−^^1^) in the column, is estimated using the expression outlined below:(5)$$q_{\mathrm{eq}(\exp)}=\frac{q_{\mathrm{total}}}{X}$$
where the total dry weight of nZVI/AC in column (g) is represented by X.

### 2.8. Analytical Methods

Around 5 mL of suspension was taken at specified intervals of time for both the batch and column experiments, filtered via a membrane filter of 0.22 μm, and acidified prior to further analysis. Using a colorimetric approach and an atomic fluorescence photometer, the concentration of Sb(V) was determined (AFS-8220, Beijing Jitian Instrument Co., Ltd., Beijing, China). ICP-MS was used to determine the net dissolved Fe concentration in the effluent of the nZVI/AC-packed columns.

## 3. Results and Discussion

### 3.1. Kinetics of Adsorption

Figure 2 clearly depicts that the kinetics of Sb(V) adsorption by NZVI/AC consists of a pair of steps: An initial quick sorption event is subsequently followed by a rather slow adsorption event. During the first 9 h, approximately 70.2% of Sb(V) was removed from simulation water, while adsorption equilibrium was reached in 72 h. As a result, in the subsequent investigations, a 72-h equilibration time was used. A nearly identical sequence was observed in the adsorption of metallic cations (e.g.,$\;{\mathrm{AsO}}_{4}^{3-}$, Mo(VI)) on NZVI/AC and the initial quick adsorption was attributed to metallic ions rapidly shifting to the outer surface of adsorbent particles, and the slow adsorption attributed to the same ions gradually diffusing within the pores existing in the intra-particle adsorbent [37].

To assess the adsorption kinetic data, we explored a variety of models, however only the intraparticle diffusion model fit well (R^2^ > 0.9) in this investigation. The data for adsorption kinetic of liquid-phase adsorption was investigated by utilizing different models to create treatment systems based on adsorption, but the intraparticle diffusion model provided the most apt explanation of the experimental data in this study (R^2^ > 0.90).

Weber and Morris demonstrated that intraparticle diffusion is the rate-limiting step in an adsorption system and that the amount of adsorbed substrate (qt) varies linearly as a function of the square root of time (t^0.5^). The adsorption speeds are then computed using this information [34]. The model is often used to analyze the control steps in the reaction and find the intra-particle diffusion rate constant of the adsorbent. If qt is plotted against t^0.5^ as a straight line and passes through the origin, it means that the diffusion in activated carbon particles is controlled by a single rate. In two steps of a plot of q_t_ vs t^0.5^, a linear relationship was discovered. As a result, Equation (1) was applied separately to each of these stages. The initial linear segment of the graph relates to adsorption on the nZVI in macropores, whereas the diffusion of Sb(V) corresponds to diffusion into micropores and/or mesopores. The nZVI particles adsorb quickly in macropores or AC channels, but their diffusion into micro-and mesopores is delayed on account of the occlusion of the majority of pores. Corrosion on the ZVI surface, adsorption in the corrosion layers, as well as diffusion all played a role. The early values for the first stage were relatively high in comparison to the second stage, precisely 5.46 times higher in simulation water, indicating that the stage 1 response had a faster velocity than the stage 2, correlating with Figure 2 and Table 2.

An identical phenomenon was seen in the process of adsorption of acidic dye, ${\mathrm{AsO}}_{4}^{3-}\;\mathrm{and}\;{\mathrm{AsO}}_{3}^{3-}$ [41], and Cu(II) and Cd(II), Mo(VI) [42,43], upon rice/modified rice husk and activated palm ash on nZVI/BC.

### 3.2. Influence of Adsorbent Dosage

The dose of the adsorbent is a significant factor in the elimination of Sb(V) via adsorption. With the increase in the dosage of nZVI/AC from 0.5 to 1.5 g·L^−1^, the elimination of Sb(V) increased from 47.3 % to 92.1 %, according to our research (see Figure 3). The clearance rate of Sb(V) rose just 1.3 % when the mass of nZVI/AC was raised from 1.5 to 2.0 g·L^−1^ (from 92.1 to 93.4 %). In all subsequent tests, the nZVI/AC dosage of 1.5 g·L^−1^ was employed. All adsorption systems attained equilibrium in 72 h, hence a 72-h equilibration time was used in all subsequent adsorption studies.

### 3.3. Influence of pH

Figure 4 illustrates the impact of solution pH on the elimination of Sb(V) by nZVI/AC. When the pH is less than 5.0, the maximal adsorption of Sb(V) occurs. The Sb(V) removal rate is 98.7% at pH 3.0. The efficiency of removal for Sb(V) decreased dramatically as the pH rose over 5.0. As the pH of the solution increases, the removal efficiency drops. The Sb(V) removal rate is only 32.7% at pH 10.5. This is acceptable since Sb(V) occurs mostly as ${\mathrm{Sb}(\mathrm{OH})}_{6}^{-}$ at pH > 3, and Sb(V) absorption is favored by positively charged surfaces. The percentage of positively charged nZVI/AC species will reduce significantly if pH is increased from 3.5 to 10.5. As a result, the amount of Sb(V) removed would decrease.

Electrostatic force discrepancies among the antimony oxyanions and sorbent surfaces existing in the solutions could explain the Sb(V) efficiency. The dominant Sb(V) solution species in the studied pH range (4–10) is ${\mathrm{Sb}(\mathrm{OH})}_{6}^{-}$ [2,13,16]. The active phase in granules has a pH_pzc_ = 7.4 [37]. More sorbent surface protonation is favored by a lower pH. The electrostatic attraction among ${\mathrm{Sb}(\mathrm{OH})}_{6}^{-}$ anions and the charged surface are enhanced by the increase in positively charged sites. Consequently, in the low pH area, the quantity of Sb(V) adsorption undergoes an increase. The adsorbent surface gets more negatively charged at high pH values. Adsorption decreases when electrostatic repulsion forces increase [13,14,15,16]. When compared to ${\mathrm{Sb}(\mathrm{OH})}_{6}^{-}$, this moderates the decline in surface repulsion as pH rises, particularly at elevated pH. As a result, the adsorption of Sb(V) on nano zero-valent iron and ferrihydrite reduced dramatically with rising pH, as has been extensively reported in earlier studies of Sb(V) adsorption upon nano zero-valent iron and ferrihydrite [2,13,15,16].

### 3.4. Effect of the Concentration of DO

Figure 5 revealed no significant effect on the removal of Sb(V) in the anoxic (DO < 0.5 mg·L^−1^) and low DO~7 mg·L^−1^ conditions, at which the removal rate of Sb(V) dropped slightly from 90.5% to 89.4%. However, the removal rate of Sb(V) dropped from ~90.1% to 61.2% under the condition of high DO > 4 mg·L^−1^, indicating that high DO has a positive effect on the removal of Sb(V) by nano-zero-valent iron. In industrial operation, the DO in raw water was usually in the range of low DO, having no significant effect on Sb(V) removal from water by nano-zerovalent iron, indicating that nZVI/AC can efficiently remove Sb(V) from water.

The results of the research showed that DO has two functions in the zero-valent iron reaction system. Moderate DO promotes the corrosion of iron and speeds up the removal of Sb(V). However, excess DO accelerates the formation of iron oxides on the surface of the zero-valent iron, causing the more serious passivation on the surface of the zero-valent iron with reduced reaction progress. The DO_H_ has been reported to have a passivation effect on zero-valent iron, which hinders the reaction of zero-valent iron [2,15].

### 3.5. Effect of Coexisting Ions

Drinking water contains a variety of anions and cations that can affect Sb(V) adsorption either positively or negatively. In the current investigation, the effects of certain common anions (${\mathrm{CO}}_{3}^{2-},C_{2}O_{4}^{2-},{\mathrm{SO}}_{4}^{2-}{,\;\mathrm{Cl}}^{-}{,\;\mathrm{PO}}_{4}^{3-}$,${\;\mathrm{SiO}}_{4}^{2-}$, ${\mathrm{CrO}}_{4}^{2-},{\mathrm{AsO}}_{4}^{3-}$) and HA on Sb(V) adsorption by nZVI/AC was thoroughly looked into (see Figure 6). The elimination of Sb was hampered by the co-existence of humic acid and anions. The co-existing anions in this study, including ${\mathrm{CO}}_{3}^{2-},C_{2}O_{4}^{2-},{\mathrm{SO}}_{4}^{2-}{\;\mathrm{and}\;\mathrm{Cl}}^{-}$, displayed a slightly negative influence on Sb(V) adsorption by nZVI/AC. However, the oxyanions phosphate, silicate, chromate, and arsenate had the most negative effects on Sb(V) elimination, whereas the HA had the least negative effect of all the oxyanions studied here [2,13,15,44]. The effect of coexisting ions on the removal of antimony particles is affected by its own concentration. On the one hand, the high concentration of carbonate ions can promote the increase of the pH of the solution, so that Sb (V) is precipitated and removed, but on the other hand, the carbonate can directly precipitate with Sb [45].

With an iron oxide surface, molybdenum, arsenate, silicate, and phosphate can create inner-sphere complexes. This would reduce molybdenum sorption as a result of their competition for identical binding sites. Sulfate ions exhibit both non-specific and specific sorption. They have a substantially poorer bonding strength with iron (hydr)oxide than Sb (V) [43].

According to the previous discussion, ${\mathrm{Sb}(\mathrm{OH})}_{6}^{-}$, deprotonating at pH > 3 is a primary regulator of Sb(V) adsorption. In aqueous media, Fe^3+^ ions are thought to form compounds with Sb(V). In consequence, the amount of dissociation/deprotonation, and overall adsorption decreases. A nearly identical phenomenon was demonstrated by the metal cation (e.g., ${\mathrm{AsO}}_{4}^{3-}$ and ${\mathrm{AsO}}_{3}^{3-}$) adsorption upon nZVI/AC in our earlier work [37,41,43].

### 3.6. Fixed-Bed Column Runs

By adjusting the 5 to 10 mL·min^−1^ flow rate with constant initial Sb(V) concentration (1.0 mg·L^−^^1^) and bed depth (20 cm), the flow rate impact on Sb(V) adsorption by nZNI/AC was examined. Figure 7 shows a graph of normalized Sb(V) concentration vs time at several flow rates.

The presence of reaction sites capable of capturing Sb(V) around or inside the adsorbents caused the adsorption to be highly rapid at low flow rates at first. Because of the gradual occupancy of these areas, the subsequent step of the process manifested in less effective uptakes. Even after breaking through, the column can accumulate Sb(V). Increased flow rates resulted in a steepening of the breakthrough curve, reducing the breakpoint time and concentration of adsorbed Sb(V). The most likely explanation is the insufficient solute residence time in the column to achieve adsorption equilibrium at that flow rate because, before being equilibrated, the Sb(V) solution departs the column. As a result, at high flow rates, the period of contact of Sb(V) with nZNI/AC is relatively brief, resulting in a decrease in absorption capacity [43].

If we increased the flow rate from 5 to 10 mL·min^−1^, the capacity for Sb(V) uptake was decreased from 6.665 to 2.433 mg, while there was an exhaustion time and earlier breakthrough in the profile. With increasing flow rate, the volume of effluent treated up to breakthrough concentration decreased, with the greatest volume reached at 10 mL·min^−1^ being 5.354 L. At the tested flow rates of 5 and 10 mL·min^−1^, the q_eq_ of nZNI/AC were 0.6275 and 0.2291 mg·g^−^^1^, respectively without WMF.

When the flow rate is 5 and 10 mL·min^−1^, the dynamic adsorption penetration curves of Sb(V) are respectively 116.4% and 144.1% when there is WMF, which compared with no WMF. nZVI accelerates corrosion by WMF, and the produced Fe^2+^ promoted the removal of Sb(V) [2,3,4,5,6,7,8,9,10,11,12,13].

### 3.7. Regeneration of NZVI/AC

Using a solution of NaOH (0.1 mol·L^−1^), the exhausted nZVI/AC was regenerated five times. Then it was eluted a few times with deionized water. According to Chinese policy, the alkaline solution was evaluated before being handed over to the special laboratory hazardous waste treatment center. Upon the regeneration of Sb(V)-saturated nZVI/AC with NaOH (0.1 mol·L^−1^), 90.0% of adsorbed Sb(V) was recovered and there was very little iron shedding [1]. The following is a description of the regeneration process.
≡Fe-Sb(V) + OH^−^ ⇄ Fe-OH + Sb(V) (*K*_des_)(6)

The antimony-loaded adsorbent was regenerated by shaking it in NaOH (0.1 mol·L^−1^) at room temperature. In 12 h, the alkaline solution desorbed 93.7% of the adsorbed Sb(V). The degradation of the produced nZVI/AC was determined by repetitive adsorption and desorption with 0.1 M NaOH. After three cycles of usage and regeneration, the adsorbent’s performance did not decline significantly, showing that the synthesized supported nano zero-valent iron was chemically and mechanically resilient for treating Sb(V)-containing drinking water. After three cycles of adsorption and desorption, the removal efficacy of Sb(V) was reduced from 93.7% to 61.9%, indicating that Sb(V) absorption into the iron (hydr)oxide structure made it difficult to elute. Similarly, following the elution of the adsorbent in backwashing operations that led to detachment of the coated iron (oxy)hydroxide, the iron-coated sand lost 13.1–20.2% of its arsenic adsorption ability [46].

### 3.8. Mechanism for Sb(V) Removal from Water by nZVI/AC

Adsorption, precipitation, and the production of Sb-Fe minerals are the key mechanisms governing the Sb(V) removal from the water via zero-valent iron. The rate of removal of Sb(V) is rather swift in the early stages of the reaction in the current study, thereby implying that adsorption is the predominant mechanism in the early phases of the reaction. Complexation or electrostatic contact could be utilized for absorptive removal of Sb(V).

During the studies, a pH of 4 was found to be the most effective pH for removing Sb(V) and to exhibit a unique behavior following the reaction. nZVI clumped together and produced bigger particles; some Fe oxides converted by corroded nZVI act as a flocculent, allowing the heavy metal to coprecipitate in water. Furthermore, the production of Fe^2+^ could hasten the transit of electrons from the Fe^0^ core, hence intensifying antimony reduction.

The loaded iron was discovered to have a substantial role in Sb(V) removal in this experiment. The nano-iron will corrode when it comes into contact with water and trace oxygen is solubilized within it [1,2,13,15]. According to the findings of EXAFS research employed by Bruce et al., nano-iron forms intermediate product species such as ferrous iron (Hydrate) oxides before producing iron (hydrated) oxides [44]. Maghemite (Fe_2_O_3_), magnetite (Fe_3_O_4_), lepidocrocite (FeOOH), and other minerals may be found in the final product. Following the above-mentioned series of multi-phase complex reactions upon the surface of iron, a multitude of hydrated oxides with high Sb(V) adsorption capability is finally generated (see Figure 8).

The expressions outlined below describe the above reactions [47]:
(1)Fe^0^ undergoes a reaction with dissolved oxygen or water to give rise to Fe^2+^:Fe^0^ + 2H_2_O → Fe^2+^ + H_2_ + 2OH^−^(7)
Fe^0^ + O_2_ + 2H_2_O → Fe^2+^ + 4OH^−^(8)(2)Further transformation of Fe^2+^ into iron (hydrated) oxides takes place by the solution pH and the oxidation-reduction potential and additional elements:6Fe^2+^ + O_2_ + 6H_2_O → 2Fe_3_O_4_(s) + 12H^+^(9)
Fe^2+^ + 2OH^−^ → 2Fe(OH)_2_(s)(10)
6Fe(OH)_2_(s) + O_2_ → 2Fe_3_O_4_(s) + 6H_2_O(11)
Fe_3_O_4_(s) + O_2_(aq) + 18H_2_O ⇌ 12Fe(OH)_3_(s)(12)

Because the surface of hydrated iron oxide contains larger active sites (-OH) which provide a high capacity for Sb(V) adsorption. The Sb(V) is thought to be adsorbed on the hydrated iron oxide surface as a bidentate binuclear chelate [14,35,43,48]. Several recent studies have shown that the surface of hydrated iron oxide adsorbs this form in neutral to weakly alkaline media. Li found an ${\mathrm{Sb}(\mathrm{OH})}_{6}^{-}$ octahedron in iron hydroxides, which verifies that Sb(V) geometry does not vary by incorporation within the corroded ZVI. Li also confirmed that Sb(V) was embedded into the iron oxide structure and that WMF only accelerated Sb(V) by ZVI [13].

## 4. Conclusions

The supported nano-iron, NZVI/AC, developed in the current work shows potential for removing Sb(V) in a neutral media amid common cations and anions found in natural water settings. Phosphate, silicate, chromate, and arsenate have been discovered to hinder the removal and adsorption of Sb(V) to varying degrees. Other frequently occurring ions, on the other hand, have minimal influence on them. The Sb(V) removal by nZVI was significantly improved with the help of WMF and was only slightly affected by co-existing anions. nZVI accelerates corrosion by WMF, and the produced Fe^2+^, in turn, promotes the removal of Sb(V). Adsorption, precipitation, and the production of Sb-Fe minerals are the key mechanisms governing the zero-valent iron-mediated removal of Sb(V) from water. The use of nZVI to remove Sb(V) with WMF could be both cost-effective and eco-friendly. However, this technology is currently only used in the laboratory, and a suitable reactor that can combine WMF and nZVI is urgently required for practical use in the future.

## Figures and Tables

**Figure 1 toxics-09-00266-f001:**
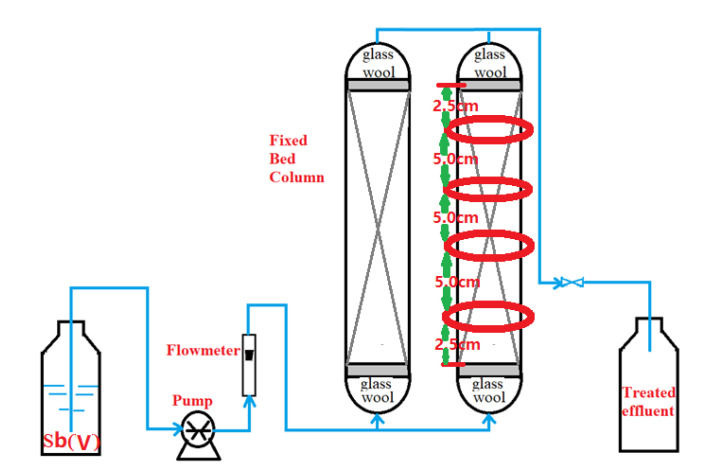
Diagrammatic illustration of the arrangement for the continuous flow system tests.

**Figure 2 toxics-09-00266-f002:**
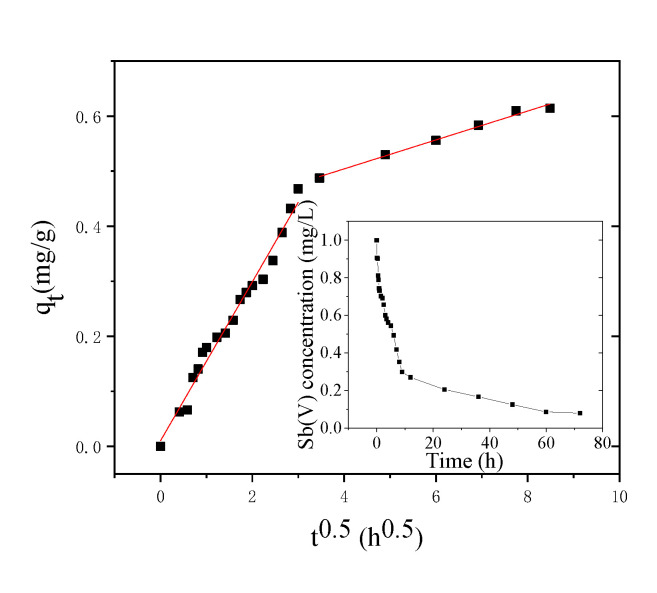
Sb(V) adsorption kinetics on NZVI/AC. The kinetics of Sb(V) adsorption onto NZVI/AC using an intraparticle diffusion model. The inset on the right depicts original data of Sb(V) adsorption as a function of time. Conditions: pH = 6.5, mesh particle size = 20 × 40, dosage of adsorbent in SW = 1.5 g·L^−^^1^, 150 rpm, T = 298 K, pH = 7, t = 72 h, C_0 Sb(V)_ = 1.0 mg·L^−1^.

**Figure 3 toxics-09-00266-f003:**
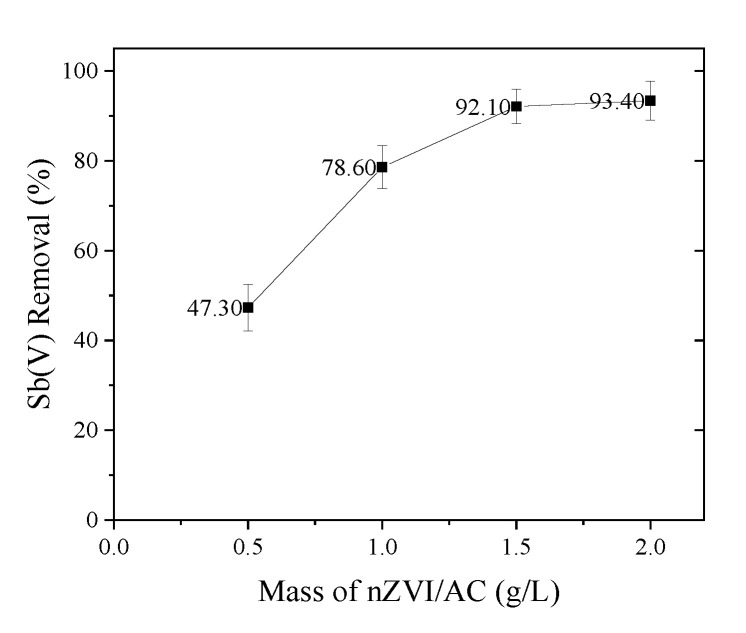
Influence of adsorbent dosage on Sb(V) adsorption on nZVI/AC. Initial [Sb(V)]: 1.0 mg·L^−1^, nZVI/AC: 0.5, 1.0, 1.5 and 2.0 g·L^−1^, pH = 7, 25 ± 1 °C.

**Figure 4 toxics-09-00266-f004:**
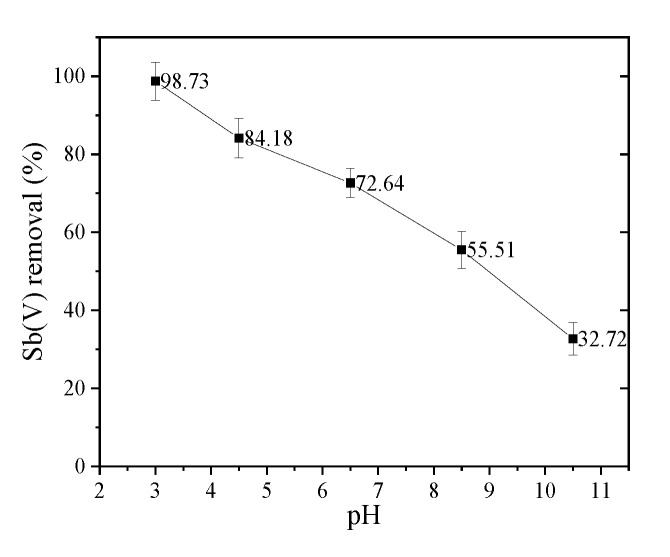
Influence of pH on Sb(V) adsorption on nZVI/AC. Initial [Sb(V)]: 1.0 mg·L^−1^, pH: 3, 4.5, 6.5, 8.5 and 10.0 nZVI/AC: 1.5 g·L^−1^, 25 ± 1 °C.

**Figure 5 toxics-09-00266-f005:**
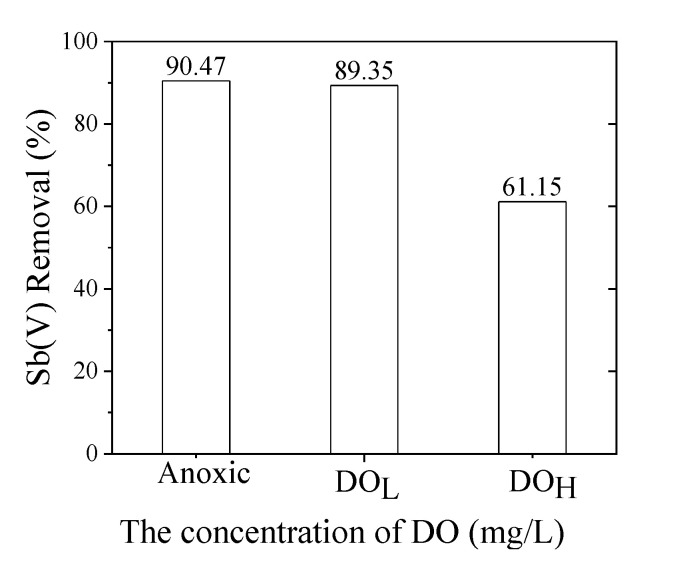
The effect of DO concentration on nZVI/AC mediated removal of Sb(V) at pH7, 25 ± 1 °C. Initial Sb(V) concentration was 1.0 mg·L^−1^; Anoxic: DO < 0.5 mg·L^−1^, DO_L_ = 7.0 mg·L^−1^, DO_H_ = 14 mg·L^−1^.

**Figure 6 toxics-09-00266-f006:**
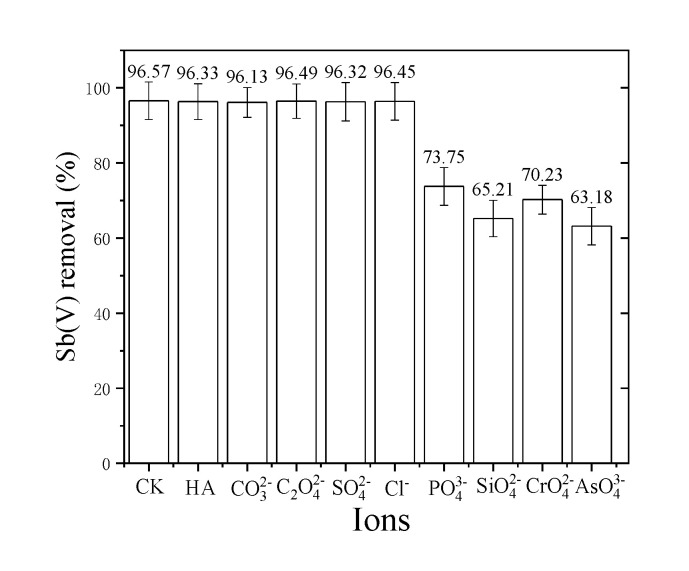
Influence of coexisting ions on Sb(V) removal by nZVI/AC at pH7, 25 ± 1 °C. Initial Sb(V) concentration was 1.0 mg·L^−1^; the molar ratio of coexisting anions to Sb(V) was 10:1; while 5 mg·L^−1^ of humic acid (HA) was added.

**Figure 7 toxics-09-00266-f007:**
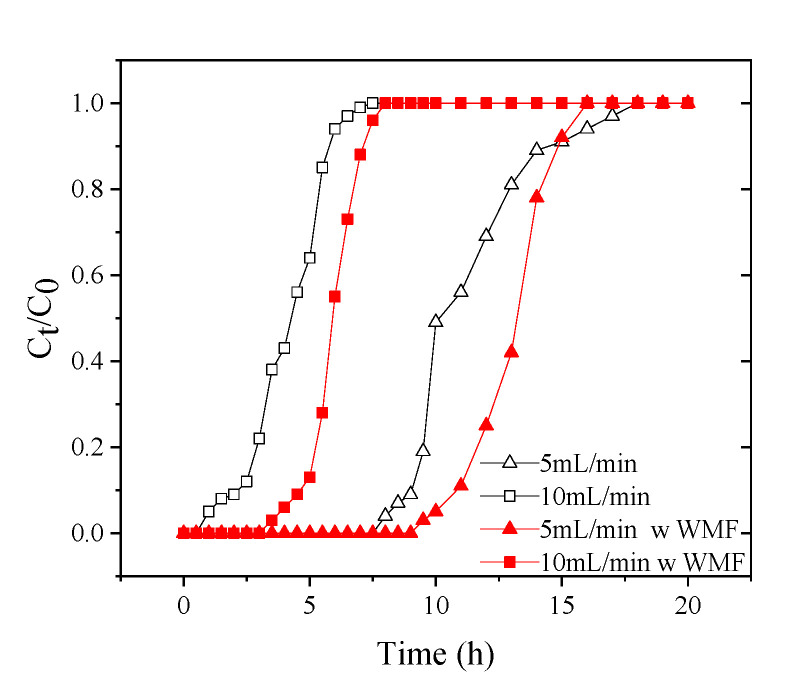
Breakthrough curves for nZNI/AC mediated Sb(V) removal at various flow rates. pH = 7, 25 ± 1 °C, H = 20 cm, C_0_ = 1 mg·L^−^^1^, flow rate = 5 and 10 mL·min^−^^1^.

**Figure 8 toxics-09-00266-f008:**
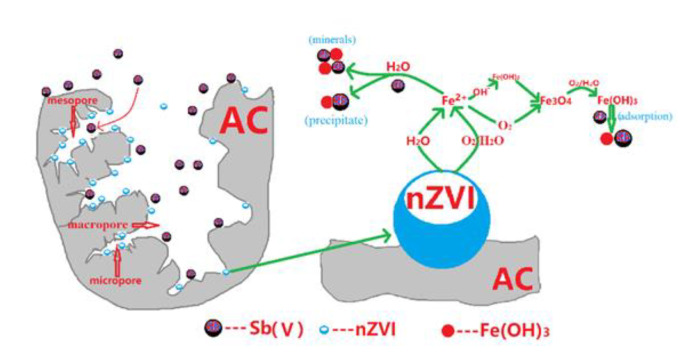
Proposed mechanism for Sb(V) removal from water by NZVI/AC.

**Table 1 toxics-09-00266-t001:** Key characteristics of nZVI/AC.

Shape	Diameter	Thickness	BET Surface Area	Fe Content	Total Pore Volume
flakes	<100 nm	~20 nm	821.7 m^2^/g	~8.2%	0.45 cm^3^/g

**Table 2 toxics-09-00266-t002:** The rate constants of adsorption kinetic model of Sb(V) on NZVI/AC.

Parameter	Weber–Morris Diffusion			
	1st Step		2nd Step	
C_0_ (1.0 mg·L^−1^)	k_id1_	R^2^	k_id2_	R^2^
Without WMF	0.1433	0.9794	0.02626	0.9866

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
