# Peer review of "Removal of Antimony(V) from Drinking Water Using nZVI/AC: Optimization of Batch and Fix Bed Conditions"

_toxics, 2021, doi:10.3390/toxics9100266_

Round 1
Reviewer 1 Report
Report on Removal of antimony(V) from drinking water using nZVI/AC: 2 Optimization of batch and fix bed conditions
toxics-1385886
Summary
Antimony (Sb) is an important resource used in various products such catalyst in plastics, flame retardants, pigments, batteries, glass, and ceramics. During mining and processing of Sb it can also be released into the environment, may cause a threat the environment but may also contaminate drinking water resources. In the present study a combination of zero valent nano iron (ZVNI) combined with activated carbon was tested regarding its ability to remove Sb from water. Thereby, Sb was supposed to be ad-or absorpt at voluminous ironoxides hydrates formed upon the reaction of ZVNI with oxygen.
For testing the new adsorbent the authors determined adsorption kinetics determined the Sb degradation at various conditions such as (pH 3-10.5, different oxygen concentrations, organic matter and inorganic ions (carbonate, chromate and dichromate, Sulfate, Phosphate, Silicate, and Arsenate).
Finally the authors postulated a mechanisms of the adsorption process.
The topic is suitable for mpdi – toxics and it is clearly written.
However, I do not recommend to publish the work in its current form do to following reasons
- I could not find any experiments with activated carbon without ZVNI. These experiments would be strongly required to see if Iron really affected the sorption process of Sb
- The influence of chloride as a most important ion in any kind of natural water, drinking water or wastewater was not tested
- The adsorption kinetics and also the adsorption capacity seemed to me quite weak. The column experiments resulted in a break-through of Sb within hours. Here a conservative inert tracer which does not adsorb at the material would help to assess how strong the columns really adsorpted Sb.
- Finally, the mechanisms seems not to be supported by the results in that the dissolved oxygen negatively affected the sorption process, while, if I understood correctly the mechanistic idea was that Sb is removed in a sorption process in the oxidized ZVNI material Line 368 f. Following the above-mentioned series of multi-phase complex reactions upon the surface of iron, a multitude of hydrated oxides with high Sb(V) adsorption capability is finally generated (see Fig.8).
Author Response
1.I could not find any experiments with activated carbon without NZVI. These experiments would be strongly required to see if Iron really affected the sorption process of Sb
Reply:
In our previous experiments, we screened common activated carbon. The activated carbon in our experiment has a weak adsorption capacity for Sb(V), only about 0.93%, so we ignored its ability to adsorb molybdenum in batches and fixed-bed.
Relevant content are marked in green in the line109.
2.The influence of chloride as a most important ion in any kind of natural water, drinking water or wastewater was not tested
Reply:
The influence of chloride on Sb(V) removal has been added.
The co-existing anions including in this study displayed a slightly negative influence on Sb(V) adsorption by nZVI/AC. However, the oxyanions phosphate, silicate, chromate, and arsenate had the most negative effects on Sb(V) elimination, whereas the HA had the least negative effect of all the oxyanions studied in this study [2, 15,44,45]. The effect of coexisting ions on the removal of antimony particles is affected by its own concentration. On the one hand, the high concentration of carbonate ions can promote the increase of the pH of the solution, so that Sb (V) is precipitated and removed, on the other hand, the carbonate can directly precipitate with Sb.[46]
Relevant contents are marked in green in the line299-307.
References:
Jialing Li, Hongliang Bao, Xinmei Xiong, Yuankui Sun, Xiaohong Guan. Effective Sb(V) immobilization from water by zero-valent iron with weak magnetic field.
3.The adsorption kinetics and also the adsorption capacity seemed to me quite weak. The column experiments resulted in a break-through of Sb within hours. Here a conservative inert tracer which does not adsorb at the material would help to assess how strong the columns really adsorpted Sb.
Reply:
Nanoscale Fe0 (nZVI) is a very promising adsorbent for Sb(V) elimination from drinking water since it possesses a high Sb(V) adsorption capacity and a large specific surface area. Moreover, nZVI has an Sb(V) removal mechanism similar to ZVI. However, the direct application of nZVI in water cleaning systems might result in nZVI losses and secondary contamination of the drinking water. Instead, nZVI supported by activated carbon (nZVI/AC) can safely and effectively remove Sb(V) due to its great porosity, low cost, and mechanical stability.
The mass of the nano zero-valent iron loading onto/into the activated carbon was 8.2 %. The activated carbon in our experiment has a weak adsorption capacity for Sb(V), only about 0.93%. Therefore, compared with nano-iron, nZVI/AC’s adsorption capacity for antimony is relatively low. However, when nano-iron is used to remove pollutants in water, if continuous production is used, a special liquid-sodium separation process needs to be added. The shortcoming of weak adsorption capacity can be compensated for its convenience of continuous cleaning water production.
Guan Xiaohong [1] adsorbed 5 mgL-1 Sb(V) with 40 μm zero-valent iron(ZVI), the adsorption capacity of ZVI is about 5 mg(Sb(V)) /g (ZVI). In our research, the adsorption capacity of nZVI/AC if calculated by nZVI, the adsorption capacity of Sb(V) is 12.04 mg(Sb(V)) /g (ZVI):
References:
[1]Effective Sb(V) immobilization from water by zero-valent iron with weak magnetic field[J]. Separation and Purification Technology, 2015, 151:276-283.
A conservative inert tracer which does not adsorb at the material does help to assess how strong the columns really adsorpted Sb. This method is often used in innovative research on analytical methods. This research focuses on potential engineering applications. We intend to apply a conservative inert tracer in the next paper. Thank you very much for your valuable comments!
4.Finally, the mechanisms seems not to be supported by the results in that the dissolved oxygen negatively affected the sorption process, while, if I understood correctly the mechanistic idea was that Sb is removed in a sorption process in the oxidized ZVNI material Line 368 f. Following the above-mentioned series of multi-phase complex reactions upon the surface of iron, a multitude of hydrated oxides with high Sb(V) adsorption capability is finally generated (see Fig.8).
Reply:
The high concentration of DO and low pH quickly passivate zero-valent iron, which has a certain inhibitory effect on the removal of As(V) and Se(VI).[1]
Oxygen has two functions in the zero-valent iron reaction system. When the oxygen content is moderate, it will promote the corrosion of iron and speed up the reaction. However, if the oxygen is excessive, it will accelerate the formation of iron oxides on the surface of the zero-valent iron, causing more serious passivation on the surface of the zero-valent iron and reducing the progress of the reaction. The high concentration of DO and low pH quickly passivate zero-valent iron, which has a certain inhibitory effect on the removal of As(V) and Se(VI) [2].
The mechanisms in Fig. 8 described the proposed mechanism for Sb(V) removal from water by NZVI/AC with low/moderate DO. The specific adsorption process is described in detail in lines 366-404
References
[1]Xuefen XIA, Yilong HUA, Xiaoyue HUANG, et al. Removal of arsenic and selenium with nanoscale zero-valent iron (nZVI)[J]. ACTA CHIMICA SINICA, 2017, 75(6): 594-601.
[2]Liang Liping. Magnetic field enhanced Se(IV) and Se(VI) removal by zero-valent iron: the efficiency and mechanism(D). 2014, Harbin Institute of Technology.

Reviewer 2 Report
Although the article has a good structure, competent construction, a fairly detailed description of the experimental methods, I would like to identify some shortcomings that, in my opinion, reduce the overall assessment:
1) 203-207: it is not shown how and with what models the study was conducted, except for the one called "the intraparticle diffusion model". Also, it is not shown how R2 calculations were performed.
2) It is not explained why the model proposed by Weber and Morris is used.
3) To explain the results obtained, only references are given without any explanations (for example, 212-reference [34]). In general, a lot of the data obtained in the work is explained only by references to previous works. The paper lacks scientific evidence made on the basis of its own research (kinetic calculations, analytical studies, etc.).
4) The conclusion refers to the economic efficiency of the proposed method, but there is no data in the paper to confirm this conclusion.
Author Response
1)203-207: it is not shown how and with what models the study was conducted, except for the one called "the intraparticle diffusion model". Also, it is not shown how R2 calculations were performed.
Reply:
The pseudo -first-order kinetic model, the pseudo -second-order kinetic model, and the intraparticle diffusion model are classic models for studying adsorption kinetics, which are mainly used to determine the rate control steps of material transfer and physical and chemical reactions in the process of adsorption kinetics. These models were tried to explain the kinetic of removing antimony from water.
R2 of the pseudo -first-order kinetic model and the pseudo -second-order kinetic model are<0.900. However, R2 of the intraparticle diffusion model is 0.9794(1st Step) and 0.9866(2nd Step). So Weber–Morris Diffusion (the intraparticle diffusion model) was conducted.
The previous results of our research team show that "the intraparticle diffusion model" can explain the kinetics of nZVI/AC to remove heavy metals (Arsenic, molybdenum, antimony) in water.
The previous research results of our research team related to the removal of heavy metals in water are as follows:
Zhu huijie, Jia Y, Wu X, et al. Removal of arsenic from water by supported nano zero-valent iron on activated carbon[J]. Journal of Hazardous Materials, 2009, 172(2-3): 1591-1596.
Zhu huijie, Shi M, Zhang X, et al. Adsorption Kinetics of Arsenic (V) on Nanoscale Zero-Valent Iron Supported by Activated Carbon[J]. Nanomaterials, 2020, 10(9): 1791.
Zhu huijie, Huang Q, Fu S, et al. Removal of Molybdenum (VI) from Raw Water Using Nano Zero-Valent Iron Supported on Activated Carbon[J]. Water, 2020, 12(11): 3162.
2) It is not explained why the model proposed by Weber and Morris is used.
Reply:
Weber and Morris have illustrated that intraparticle diffusion is the rate-limiting factor in an adsorption system and the mass of the adsorbed substrate changes linearly as a function of the square root of time (t1/2). These data were then used to calculate the speeds of adsorption. When qt was plotted as a function of t1/2 a linear relationship was seen in two separate stages (Fig. 2b). Therefore, Eq. (1) was applied to these two stages separately. The first linear section of this graph corresponded to the adsorption on the NZVI located inside the macropores while the second very likely corresponded to the diffusion of Sb(V) into micro- and/or mesopores. Adsorption of Sb(V) on the NZVI particles located inside the AC macropores or -channels was quick, while their diffusion into micro- and mesopores was slow because most of these pores were blocked. Apart from the pore diffusion process, the corrosion of the ZVI surface, adsorption, and diffusion in the corrosion layers were also involved.
The kinetics of arsenic/molybdenum/antimony adsorption by NZVI/AC included two steps: a fast initial sorption followed by a much slower sorption process. It can be explained with Weber and Morris model.
The previous results of our research team show that the Weber and Morris model can explain the kinetics of nZVI/AC to remove heavy metals (Arsenic, molybdenum, antimony) in water.
3) To explain the results obtained, only references are given without any explanations (for example, 212-reference [34]). In general, a lot of the data obtained in the work is explained only by references to previous works. The paper lacks scientific evidence made on the basis of its own research (kinetic calculations, analytical studies, etc.).
Reply:
The references are given explanation.” The model is often used to analyze the control steps in the reaction and find the intra-particle diffusion rate constant of the adsorbent. If qt is plotted against t0.5 as a straight line and passes through the origin, it means that the diffusion in activated carbon particles is controlled by a single rate.”
Relevant content are marked in green in the lines 217-220.
Our research team characterized the nature of adsorbed arsenate on ferrihydrite with Infrared spectroscopic and X-ray diffraction(please see reference [1]).
The following is the research we have done before on nZVI/AC(please see reference [2]):
(a) 2,000times
(b) 20,000 times
(c) 3,5000times
Fig. 1 SEM image of the activated carbon supported nanoscale zero-valent iron
Fig. 2 Nitrogen adsorption isotherms on activated carbon:AC and NZVI/AC at 77 K
References
[1] Y. F. Jia, L. Xu, X. Wang, G. P. Demopoulos, Infrared spectroscopic and X-ray diffraction characterization of the nature of adsorbed arsenate on ferrihydrite. Geochim. Cosmochim. Acta. 71 (2007) 1643-1654.
[2] Zhu huijie, Jia Y, Wu X, et al. Removal of arsenic from water by supported nano zero-valent iron on activated carbon[J]. Journal of Hazardous Materials, 2009, 172(2-3): 1591-1596.
4) The conclusion refers to the economic efficiency of the proposed method, but there is no data in the paper to confirm this conclusion.
Reply:
The method is cost-effective because (1) the removal of heavy metals such as antimony in the water can be operated continuously without special solid-liquid separation, (2) the adsorbent can be regenerated and reused, (3) the waste adsorbent does not need to be specially treated, and the cost of iron and activated carbon used is low. Overall, it is economical and efficient. Because there are many reports on this in the literature, there is not much explanation and data in this study.

Reviewer 3 Report
The authors conducted well-designed research focused on the removal of antimony(V) from water by the action of nZVI/AC. The goal of the research has been clearly defined and the presented results may be interesting for the readers of the Journal. The introduction determines the purpose of the work and its significance. The structure of the manuscript is clear, the Figures and Tables, in general, are correct and illustrative, and also the number of references handled is good. The conclusions fully summarize the most important aspects of the work. Nevertheless, I suggest improving some elements of the paper:
- Please explain the abbreviation DO in the abstract and in the main text.
- The magnetic rings on Fig.1 should be labeled. How many magnetic rings were used?
- The resolution of Figures 2 and 7 should be improved.
- The removal rates should be presented to one decimal place.
Author Response
1.Please explain the abbreviation DO in the abstract and in the main text.
Reply:
DO is an abbreviation for dissolved oxygen.
Relevant content are marked in green in the line 20 and 97.
2.The magnetic rings on Fig.1 should be labeled. How many magnetic rings were used?
Reply:
Two parallel continuous column systems possessing a height of 200 mm,, with a ring magnet every 5.0 cm. There are 4 magnetic rings in total.200mm=20.0cm, 20.0 cm÷5.0 cm =4(magnetic rings).
The space between the two rings is shown in Figure 1.
3.The resolution of Figures 2 and 7 should be improved.
Reply:
The resolution of Figures 2 and 7 have been improved.
4.The removal rates should be presented to one decimal place.
Reply:
The removal rate has been modified to one decimal place in the line 278, 279 ,351 and 364.

Round 2
Reviewer 1 Report
The manuscript significantly improved and I have no further comments